# Genetic Landscape of Rett Syndrome Spectrum: Improvements and Challenges

**DOI:** 10.3390/ijms20163925

**Published:** 2019-08-12

**Authors:** Silvia Vidal, Clara Xiol, Ainhoa Pascual-Alonso, M. O’Callaghan, Mercè Pineda, Judith Armstrong

**Affiliations:** 1Sant Joan de Déu Research Foundation, 08950 Barcelona, Spain; 2Institut de Recerca Pediàtrica Hospital Sant Joan de Déu, 08950 Barcelona, Spain; 3Neurology Service, Hospital Sant Joan de Déu, 08950 Barcelona, Spain; 4CIBER-ER (Biomedical Network Research Center for Rare Diseases), Institute of Health Carlos III (ISCIII), 28029 Madrid, Spain; 5Molecular and Genetics Medicine Section, Hospital Sant Joan de Déu, 08950 Barcelona, Spain

**Keywords:** Rett syndrome, Rett-like, NGS, genetics

## Abstract

Rett syndrome (RTT) is an early-onset neurodevelopmental disorder that primarily affects females, resulting in severe cognitive and physical disabilities, and is one of the most prevalent causes of intellectual disability in females. More than fifty years after the first publication on Rett syndrome, and almost two decades since the first report linking RTT to the *MECP2* gene, the research community’s effort is focused on obtaining a better understanding of the genetics and the complex biology of RTT and Rett-like phenotypes without *MECP2* mutations. Herein, we review the current molecular genetic studies, which investigate the genetic causes of RTT or Rett-like phenotypes which overlap with other genetic disorders and document the swift evolution of the techniques and methodologies employed. This review also underlines the clinical and genetic heterogeneity of the Rett syndrome spectrum and provides an overview of the RTT-related genes described to date, many of which are involved in epigenetic gene regulation, neurotransmitter action or RNA transcription/translation. Finally, it discusses the importance of including both phenotypic and genetic diagnosis to provide proper genetic counselling from a patient’s perspective and the appropriate treatment.

## 1. Introduction

Rett syndrome (OMIM#312750) is an early-onset neurodevelopmental disorder, which was first described by Doctor Andreas Rett in 1966 [1]. However, it was not until 1999 when Zoghbi’s laboratory identified mutations in the X-linked methyl-CpG-binding protein 2 gene (*MECP2*; OMIM*300005) in RTT patients. This gene encodes a chromatin-associated protein that contains a methyl-CpG binding domain and can activate and repress transcription; it is essential for the maturation of neurons and normal function of nerves cells [2]. The molecular pathogenesis of *MECP2* mutations is complex, involving multiple functions and tissues. MeCP2 has two differentially spliced isoforms of exons 1 and 2 (MeCP2-e1 and MeCP2-e2) which contribute to the diverse functions of MeCP2, but only mutations in exon 1, not exon 2, are observed in RTT. However, the majority of RTT mutations occur in *MECP2* exons 3 and 4 [3]. MeCP2-e1 contains three exons and the start codon is located in exon 1, while MeCP2-e2 contains exons 2, 3 and 4 and the start codon is located in exon 2 [4]. Moreover, MeCP2-e1 is mainly expressed in the central nervous system [3], suggesting that it is the dominant isoform in the brain [5]. Studies using mice models have shown that *Mecp2-e1* deficiency alone contributes to neurologic symptoms, while *Mecp2-e2* has no essential function in the nervous system [5]. More than 800 different mutations in *MECP2* have been identified in more than 95% of patients with classic RTT and 75% of patients with atypical RTT (RettBASE: MECP2 Variation Database: http://mecp2.chw.edu.au/) [6,7]. There are also some atypical RTT variants, such as the early onset seizure variant and the congenital variant, which have been associated with mutations in cyclin-dependent kinase-like 5 (CDKL5; Xp22; OMIM*300203) and forkhead box protein G1 (FOXG1; 14q12; OMIM*164874), respectively [8,9]. However, the etiology of a subset of patients with a clinical diagnosis of RTT still remains unknown.

Nowadays, with the increasing use of Next-Generation Sequencing (NGS) techniques, and the improvement of the techniques themselves and bioinformatics analysis tools, more patients can obtain a genetic diagnosis, which is important for proper genetic counselling, the patient’s future perspective, and treatment options. Consequently, the number of known genes which are disease-causing for RTT-like phenotypes increased remarkably in the last years. This development can actually be observed in the hugely heterogeneous group of neurodevelopmental disorders [10]. This study underlines the current molecular genetic studies performed in RTT patients, highlights the phenotype overlap with other monogenic disorders, and reviews the new treatments that are being performed.

## 2. RTT and RTT-Like Syndrome

The diagnosis criteria used to establish the clinical diagnosis of RTT was described in 2002 and revised in 2010 [11]. The patients diagnosed with classical RTT should present four main criteria: Partial or complete loss of spoken language, partial or complete loss of purposeful hand movements, gait abnormality, and stereotypic hand movements. In contrast to classical RTT, a diagnosis of atypical RTT is described when the patient present two or more of the main criteria in addition to five or more of the supportive criteria. Nearly all classical RTT patients are characterized by a period of apparently normal development followed by a regression phase, but some atypical forms are congenital and early seizure shows developmental impairment/delay from the first months of life. In addition, there are some clinical features that can exclude a diagnosis of classical RTT, such as brain injury, a neurometabolic disease, or neurological infection [12]. Nowadays, in RTT, as in other diseases, the term “like” is used in patients that do not fulfill established clinical criteria, but present an overlapping phenotype with the disease. Formal consensus criteria for a Rett-like syndrome (RTT-like) are not published yet and a combination of distinct features of RTT can be described as RTT-like phenotype. 

The report of pathogenic or likely pathogenic variants in different genes in patients with overlapping phenotypes creates a huge challenge in the clinical diagnosis. NGS, such as gene panels or whole exome/genome sequencing, allows us to solve difficulties and improves results, complementing the clinical diagnosis with a genetics diagnosis. However, making an accurate phenotypic description of the patients is crucial to enable the selection of the most relevant genes to be analyzed and for assessing the clinical significance of genomic variants identified in them.

## 3. New Technologies for a Rare Genetic Diagnosis 

In recent years, NGS technology—a method of simultaneously sequencing millions of fragments of DNA—has emerged as a powerful tool for the study of this type of genetic disease. Now, with the possibility of multiplexing genes and patients, sequencing them at the same time, the cost-efficiency of the technique is comparable to the Sanger sequencing analysis of a single gene [13]. Therefore, the global implementation of these technologies in research laboratories has led to an important increase in the identification of diseases or genes related to RTT/RTT-like phenotype that in some cases had previously been associated with other well-described diseases [14,15,16]. While the added value of NGS diagnostics in all of these patients is clear, an optimal implementation strategy for diagnostic laboratories is yet to be established [17].

Basically, there are three NGS approaches for DNA sequencing which can be used to improve the diagnostic rate in this hugely heterogeneous group of diseases: (1) Targeted enrichment of a set of genes (gene panels); (2) whole-exome sequencing (WES); (3) whole-genome sequencing (WGS). Targeted panels focus on individual genes, specific regions of interest, or a subset of genes associated with a wide variety of inherited disorders. This approach is usually the first line of testing, while WES is reserved for cases in which targeted testing has been uninformative [18]. Moreover, panels can be customized and optimized for different regions and sample types, allowing determination of single nucleotide variants (SNVs) from NGS in a more cost-effective manner. The targeted panels are constantly improving because with basic research and WES and WGS of patients without a genetic diagnosis, new genes are discovered or their functions are more clearly understood and, subsequently, are associated with human diseases. For this reason, targeted panels are the best approach in terms of genetics diagnosis.

WES testing often involves testing the child and both parents (trio testing) to assist in the interpretation of variants [19,20]. The current challenge of WES is to determine benign variants from pathological variants, since the exome of a healthy person reveals about 30,000–100,000 variations if compared with the standard reference genome. Using variation databases and software tools, such as the ones that are described in Table 1, the potential disease-causing variation can be detected, although the exact method is uncertain and functional studies continue to be necessary to fully demonstrate the pathogenicity of the variations found [21]. However, mutations in intronic and promoter regions are not covered and nor are greater structural variants like inversions and translocations.

Nowadays, WGS is considered to be the most comprehensive genetic test available, but it is not applied to patient diagnostics because of the complex and challenging data analysis, the high cost compared to targeted panels and WES, and the unknown diagnosis potential of the test. With the sequencing of the whole genome we can detect, besides from SNV in coding regions such as WES, variations in non-coding regions, but in the majority of cases functional studies are required to determine their pathogenicity. To date, there are no publications about WGS in a cohort of RTT nor RTT-like, but studies in intellectual disability (ID) or other diseases are published [22,23]. However, targeted panels have the best cost-efficiency value. 

Moreover, NGS not only allows for the detection of SNV variants but with this technology we are also able to detect copy number variations (CNVs); the size and complexity of the genomic regions of interest will determine which NGS method is the best to use. WGS offers the potential to capture all genetic variations, including CNVs and structural changes such as inversions and translocations. WES can detect indels (CNVs ~1–100 bp), but approaches are steadily improving to provide data suitable for larger CNVs because WES covers many different regions and it can be particularly tricky to optimize for uniformity of coverage. Finally, targeted NGS panels offer high uniformity of coverage of targeted regions, and the targeted nature results in lower costs with increasing depth, opening up the possibility for reliable CNV calling.

Recent studies have shown that germline and somatic mosaicism is present in genes related with severe encephalopathies such as Dravet syndrome [24,25], focal cortical dysplasia [26] and intellectual disability [27]. Mosaicism is being postulated as the cause to explain differential phenotype expression of the disease among patients (somatic mosaicism due to a postzygotic mutation) and to explain recurrent mutations in the same family assumed de novo (due to low-grade parental mosaicism). To study mosaicism, other techniques besides NGS must be considered, such as single-molecule molecular inversion probes (smMIP), a technique with high sensitivity for detecting low-grade mosaic variants. Using this technique, it has been shown that parental mosaicism occurs in a substantial proportion of families with mutations in *SCN1A* (7%–13%), which has important implications for recurrence risks in subsequent pregnancies. On the other hand, to study disease severity ranges between patients with mutations in the same gene, a better prediction technique is needed because Dravet syndrome patients with mosaicism have milder phenotypes than those with heterozygous mutations [28].

## 4. NGS Results: Many Genes, Many Disorders

Apart from RTT and RTT-like patients, which have mutations in the *MEPC2, CDKL5* and *FOXG1* genes, there are a percentage of patients without a genetic diagnosis. Now, with WES and panels that incorporate more and more genes related to the central nervous system, the number of patients with a pathogenic variant detected has increased in genes that were previously not related to RTT nor RTT-like. All studies published about these genes are summarized in Table 2.

Using NGS, in only five years more than eighty genes were related to the RTT-like phenotype and some of these genes were identified as causative for aRTT or RTT-like phenotype in these patients, although some of them were associated with well-known syndromes such as Pitt–Hopkins syndrome (*TCF4*, *CNTNAP2* and *NRXN1* genes), Phelan–McDermid syndrome (*SHANK3* gene), Angelman syndrome (*UBE3A* gene), Kleefstra syndrome (*EHMT1*) and Cornelia de Lange syndrome (*SMC1A*). In addition, a substantial number of genes are epileptic encephalopathy genes (*STXBP1*, *SCN1A*, *SCN2A*, *SCN8A*, *GRIN2A*, *GRIN2B*, *HCN1*, *SLC6A1*, *KCNA2*, *EEF1A2*, *KCNB1* and *SYNGAP1*) and are related with mental retardation and epilepsy (*IQSEC2* and *MEF2C*). 

Note that, regarding the number of patients described in these publications, there are some genes that are more represented (Figure 1). *STXBP1, TCF4, SCN2A, WDR45,* and *MEF2C* are the most common genes with a pathogenic (or likely pathogenic) variant detected in patients with RTT/RTT-like phenotype. 

The *STXBP1* gene (syntaxin-binding protein 1; OMIM#602926) encodes a transmembrane attachment protein receptor that plays an important role in the release of neurotransmitters via regulation of syntaxin, a transmembrane attachment protein receptor [47]. Pathogenic variants in this gene, reducing its expression, depress the functions of GABAergic and glutamatergic synapses, particularly in GABAergic interneurons [48], a process that has been shown to be altered in RTT patients [49]. Mutations in *STXBP1* have been associated with epileptic encephalopathy, early infantile 4 (EEIE4) and a series of neurodevelopmental disorders, including RTT-like syndrome. 

The *TCF4* gene (transcription factor 4; OMIM*602272) is a broadly expressed basic helix–loop–helix (bHLH) protein that functions as a homodimer or as a heterodimer with other bHLH proteins. These dimers bind DNA at Ephrussi (E) box sequences motif (‘CANNTG’). Alternative splicing creates numerous N-terminally distinct TCF4 isoforms that differ in their subcellular localization and transactivational capacity [50]. Mutations in *TCF4* have been associated with Pitt–Hopkins syndrome (PTHS), which is characterized by severe ID, delayed motor development, seizures, wide mouth and distinctive facial features, hypotonia, microcephaly, limited walking abilities, and intermittent hyperventilation followed by apnea [31]. The microcephaly, intermittent hyperventilation, and stereotype hand movements may steer clinicians towards a diagnosis of RTT-like rather than PTHS. The distinct facial features presented in patients with a clear PTHS phenotype, such as thin eyebrows, sunken eyes, a pronounced double curve of the upper lip, and a wide mouth with full lips, is more consistent with PTHS and helps to distinguish PTHS from RTT, but some patients do not always present distinctly these facial dismorphisms, or they are often not well-defined during the first year of life [51]. 

*SCN2A* (neuronal voltage-gated sodium channel NaV1.2; OMIM*182390) encodes one member of the sodium channel alpha subunit gene family, responsible for generation and propagation of action potentials, chiefly in nerve and muscle. Pathogenic variants in the *SCN2A* gene that produce loss-of-function of the protein lead to ASD/ID and increased channel activity that lead to epileptic encephalopathy early infantile 11 (EEIE11) and benign familial neonatal-infantile seizures (BFIS) [52].

WD40 repeat proteins are an important key component that regulates the assembly of multiprotein complexes by presenting a beta-propeller platform for simultaneous and reversible protein–protein interactions [53]. Variants in *WDR45* (WD repeat-containing protein 45; OMIM*300526) are associated with developmental delay in early childhood and progressive neurodegeneration in adolescence or adulthood related to iron accumulation in the *globus pallidus* and *substantia nigra* [30,54]. Affected patients may have features overlapping those of RTT, including developmental regression, hand-wringing, and seizures. Some may even have a diagnosis of typical or atypical RTT [55].

*MEF2C* (OMIM*600662) belongs to the myocyte enhancer factor-2 (MEF2) family of transcription factors. *MEF2C* plays an important role in myogenesis, development of the anterior heart field, neural crest and craniofacial development, and neurogenesis, among others [56]. It is well-described the *MEF2C* haploinsufficiency syndrome that has been recognized as a neurodevelopmental disorder. Until now, fourteen patients with point mutation pathogenic, or likely pathogenic, variants in *MEF2C* have been identified in RTT-like patients, including three nonsense, three missense and three frameshift variants [37,57].

In Table 3, the most common genes for RTT-like phenotypes are summarized, and all of them present an autosomal dominant inheritance pattern, either autosomal or linked to the X chromosome and are caused by de novo heterozygous mutations in the germline. Note that all the diseases are severe IDs and most of them meet the four main criteria for RTT; all of them present a loss or speech severe deficit, gait abnormalities and lost or absent purposeful hand movements linked to stereotypical hand movements, such as hand wringing. Only developmental regression, one of the four main criteria, is absent in PTHS. Moreover, other common symptoms in RTT are present in these diseases too, such as epilepsy, breathing disturbances and microcephaly, although not all the RTT patients present them. In contrast, other symptoms that are not present in RTT are present in atypical forms related to *CDKL5* and *FOXG1*, such as dysmorphic facial features in PTHS or CNS abnormalities in neurodegeneration with brain iron accumulation.

Focusing on the pathophysiology of RTT at the brain level, female mice heterozygous for the null *MECP2* present microcephaly without gross neuropathological changes. Specifically, mouse and human neurons without MeCP2 have smaller somas and decreased dendritic complexity [58,59]. A decrease in synaptic plasticity and abnormalities in neurotransmitter concentrations is also observed in many neuronal types [60,61,62]. Looking at the established knockout mice models of those of the most common genes detected in RTT patients, morphological and/or physiological features resemble RTT mouse models. For instance, young adult *Mef2c* and *Scn2a* cKO mice present a normal gross brain morphology and cortical layer organization, as murine RTT models. In contrast, *Mef2c* cKO mice cause an increase in dendritic spine density on dentate granule neurons of the hippocampal dentate gyrus [63] and *Scn2a* cOK in pre-oligodendrocyte alters their morphology, impairs myelination and reduces axon-oligodendrocyte interactions [64,65]. Moreover, depletion of specific presynaptic proteins involved in exocytosis, such as *STXBP1*, causes abnormalities in neurotransmitter concentrations and produces neuronal cell death [66]. *TCF4* haploinsufficiency mice exhibit a delay in neuronal migration, and a significant increase in the number of upper-layer cortical neurons, as well as abnormal dendrite and synapse formation [67]. *WDR45* cOK mice show a loss of neurons in prefrontal cortex and basal ganglion in aged mice and increased apoptosis in prefrontal cortex, recapitulating a hallmark of neurodegeneration [68]. 

At the moment, more than 80 genes have been associated with RTT/RTT-like, but their link with RTT should be critically evaluated. In order to relate a gene with RTT several points have to be taken into consideration: (1) Proper clinical characterizations of the patients; (2) deep comprehension of the functions of the candidate gene; (3) validated evidence that mutation found it is pathogenic. The more patients with a similar phenotype have pathogenic mutations in the same gene, the more consistent evidence we will have of that gene being part of the list of genes related to RTT. 

## 5. Functions and Pathways around RTT

The list of genes related to RTT/RTT-like phenotypes is complex and diverse. However, using REACTOME (an open-source, open access, manually curated and peer-reviewed pathway database; https://reactome.org) and STRING (a database of known and predicted protein–protein interactions; https://string-db.org/), we can identify some groups of genes with functions involved in common mechanisms. Several pathways can be studied in RTT/RTT-like patients, such as chromatin modulation, synaptic function and ubiquitin conjugation (Figure 2).

This list currently includes 15 genes (*ACTL6B, ANKRD31, CHD4, HDAC1, JMJD1C, MEF2C, NCOR2, SATB2, SMARCA1, TBL1XR1, TRRAP, ZFX. ZNF238, ZNF620* and *ZSCAN12*) involved in chromatin modulation pathways, such as chromatin-modifying enzymes and histone deacetylases (HDACs) and 21 genes (*ATP6V0A1, CACNA1I, CHRNA5, GABBR2, GABRB2, GABRD, GRIN2A, GRIN2B, HCN1, IQSEC2, KCNA2, KCNJ10, KCNQ2, SCG2, SCN1A, SCN2A, SCN8A, SHANK3, SLC6A1, STXBP1* and *SYNGAP1*) involved in synaptic function, necessary for GABAergic, glutamatergic and dopaminergic synapses, synaptic vesicles trafficking, ion homeostasis in neurons and circadian entrainment. The link of all these genes to the same pathways could explain why these patients’ phenotypes overlap, causing impaired synaptic function, sleep disturbances and major dysregulation of gene expression. Notably, there are also a few genes (*MGRN1, RHOBTB2* and *USP8*) involved in ubiquitination processes, which *UBE3A* (the gene responsible for Angelman syndrome) is also linked to. The considerable overlapping of RTT and Angelman features could be explained due to this relationship.

MeCP2 performs many tasks during the neurodevelopment, such as regulating the gene expression of other genes, modulating epigenetic imprinting and neurotransmitter actions. Hence it is challenging to create a well-defined pathways involving MeCP2 and predict the downstream effects that disruption of the MeCP2 function can generate. Ehrhart et al. 2016 [69] created a comprehensive visualization of the biologic pathways showing how *MECP2* upstream and downstream regulation developed. Moreover, it had been and published on WikiPathways which will serve as template for future omics data driven research (http://www.wikipathways.org/instance/WP3584) [69]. 

Discovering the pathways related to *MEPC2*, we could better link genes that are mutated in patients without *MECP2* defects. Nowadays, the combination of the omics data analysis and prior knowledge databases are a powerful approach to identify connections between mutation and phenotype. Ehrhart et al. 2019 [70] identified a subset of genes, which are significantly different in several transcriptomics datasets and were not described yet in the context of RTT. They described that these genes are involved in molecular pathways and several processes known to be affected in RTT patients [70]. For example, altered calcium homeostasis seems to be responsible for an abnormal neuronal development and generates epilepsy; and tubulin, *ERM* and *MEF2C* are some of the altered proteins related to cytoskeletal abnormalities that are present not only in RTT, but also in Angelman syndrome [69]. In the same way, cholesterol biosynthesis is altered in RTT and in Smith–Lemli–Opitz syndrome, in which it has been pointed out to be the cause of the autism and malformations [71]. The NF-kB pathway, which is involved in nervous system development, synaptic transmission and cognition, is altered in RTT and RTT-like patients and seems to be the cause of mental retardation [72]. Another pathway observed in patients with RTT, autism, and Parkinson’s disease (PD) is the neurotransmitter imbalance of GABA, Glutamine and Dopamine. This imbalance seems to be responsible for the autism features RTT patients present and the motor difficulties that patients with PD and RTT have [73].

An important feature of MeCP2 reduction in RTT mouse models and individuals with RTT is a propensity for seizure, a prominent signature in many brain diseases, including RTT [74,75]. The deletion of MeCP2 from all forebrain GABAergic interneurons recapitulates major phenotype of RTT [75], demonstrating that altered inhibitory function is critical for normal function of GABA-releasing neurons and an important aspect of RTT pathophysiology. It has been demonstrated recently that human neurons derived from patients with RTT and RTT mouse models show a significant reduced the *SLC12A5* gene expression, resulting in a delayed GABA functional switch [76]. This gene encodes a neuron-specific K+/Cl− cotransporter 2, the major extruder of intracellular chloride in mature neurons. Moreover, it has been established that MeCP2 regulates KCC2 expression through inhibiting RE1-silencing transcriptional factor [77], and it is suggested that KCC2 should play a role in the pathophysiology of RTT

Recently, Cosentino et al. (2019) have described several alterations in RTT patients and animal models during the pre-symptomatic stage [78]. During this stage some compensatory mechanisms keep the phenotypic outcome to a minimum until *MECP2* deficiency cannot be supplied and the known phenotype of RTT becomes apparent [79]. Thus, since the alterations found in RTT spectrum disorders are due to both direct and indirect effects of *MECP2* and related genes’ deficiencies/malfunctions, the best approach for characterization and consequent treatment would be the comprehensive study of all the altered pathways that have been discovered in these patients. If we were able to find a way to compensate those altered pathways, a treatment could be implemented in very early stages of development, even before the onset of the most remarkable features of the syndrome [78,79].

## 6. Future Perspectives and Treatment Options

In recent years, impressive advances have been achieved not only in the genetics diagnosis of neurodevelopmental disorders, but also in elucidating the physiological pathway and molecular mechanisms involved in the clinical manifestations of the diseases. An increasing amount of evidence is emerging that understanding these mechanisms is relevant for the selection of the most appropriate treatment in the affected individual. 

More than 50 years have passed since the description and clinical characterization of RTT and the current standard of care for patients remains limited to supportive and symptomatic therapies that can palliate the symptomatology of the patients, but not cure the disease per se. Drug treatment consists mainly of off-label prescriptions due to the lack of approved medications for the disorder. Until now, all RTT trials have been based on the essential role of *MECP2* in the development and maintenance of neurons in the central nervous system, and its specific role in the distinct cellular subtypes focused on the following specific neurotransmitters: dextromethorphan (an NMDA receptor antagonist, mainly used for cognition and seizures [80]); desipramine (a noradrenaline reuptake inhibitor, for breathing abnormalities [81]); and IGF-1 [82]. To date, emulating the MeCP2 function, using a pharmacology strategy, as a treatment for RTT is not the best and most successful strategy. 

However, it is not only patients with RTT or RTT-like that are caused by defects in the *MECP2* gene. An improved understanding of the different genes mutated in the RTT-like phenotype generates important therapeutic clues and opportunities to develop novel and better treatments. A treatment which targets neuronal maturational defects seen in *MECP2* mutations may not be effective for an ion channelopathy due to *KCNB1* alterations. In this way, therapies must focus on personalized treatments for each individual, depending on which gene and which type of mutation carries. For example, for the *SCN2A* gene that encodes the voltage-gated sodium channel Nav1.2, gain-of-function versus loss-of-function variants in *SCN2A* determine whether sodium channel blockers improve or worsen seizure control [52]. Another example is that L-serine supplementation might ameliorate *GRIN2B*-related severe encephalopathy [83]. Additionally, for *STXBP1*, which plays an important role in presynaptic vesicle docking and fusion [84], current treatments are largely limited to seizure control and future therapies will also need to target the developmental aspects of the disease [83]. It has been hypothesized that enhancing *KCC2* expression could increase the efficacy of GABAergic inhibition and also improve the dendritic spine and excitatory synapse development, both of which are abnormal in RTT [85,86] and may be caused by an aberrant interaction between *KCC2* and the dendritic cytoskeleton [87,88]. Recently, two works have considered the pathogenic role of diminished KCC2 expression in the Mecp2 null model: The first found that the use of bumetanide can attenuate the unbalance glutamatergic/GABAergic ratio if treated in the early stages of the disorder [89]; and Tang et al. (2019) have shown that the injection of KEEC KW-2449 or piperine (small-compounds) in Mecp2 mutant mice ameliorated disease-associated respiratory and locomotion phenotypes [90].

## 7. Conclusions

All of the individuals summarized in this review met the diagnostic criteria for RTT or RTT-like; however, lacking a defect in *MECP2* underscores the importance of carrying out additional genetic testing, whether it is by specific gene panels, WES or WGS, to identify the specific etiology and to direct appropriate diagnostic and therapeutic strategies. Many disorders can be caused by multiple genes, such as West syndrome or Charcot-Marie-Tooth, and these are considered as the same disorder as long as they share a common phenotype. Therefore, we consider that all classical, atypical and all RTT-like phenotypes could be grouped into an RTT spectrum disorder with many causative genes.

From a research point of view, the new genes recently associated with RTT-like phenotypes without a clear description of their biological functions network involved need further study. Furthermore, variants with an unknown clinical significance in genes without clear defined functions have been found through screening RTT spectrum cohorts without genetic diagnosis. These new candidate genes need functional studies to establish their potential role in the disease pathogenesis. From a clinical perspective, a better definition of the pathways that connect of all these genes involved in all RTT and RTT-like phenotypes would enable a better understanding of the genetic landscape of the RTT spectrum. The elucidation of the functional pathways involved in all these patients could support the development of future targeted therapies. 

## Figures and Tables

**Figure 1 ijms-20-03925-f001:**
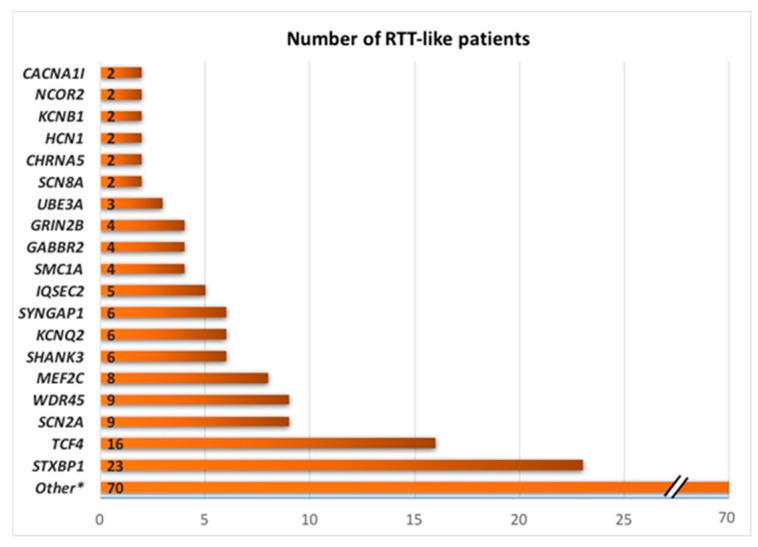
Overall results of all the summarized genetic studies. Patients grouped by gene defect.

**Figure 2 ijms-20-03925-f002:**
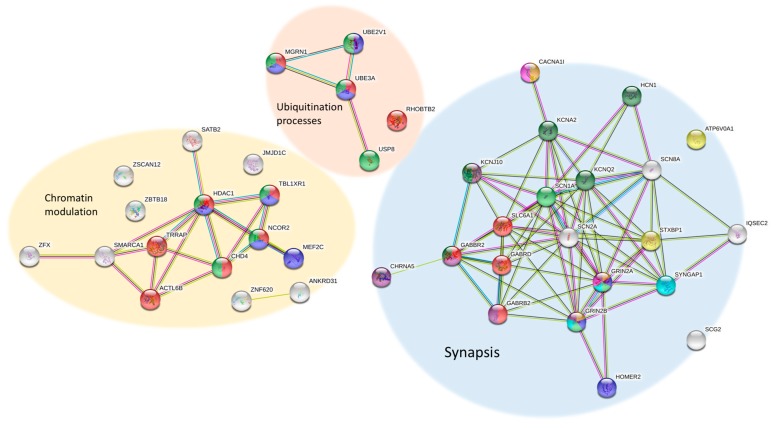
RTT-related protein known functional interaction networks. (A) Chromatin modulation: Red for chromatin modifying enzymes, blue for proteins involved in MECP2-mediated transcriptional regulation and green for histone deacetylases (HDACs). (B) Ubiquitination processes: Red for proteins linked to ubiquitin-mediated proteolysis, green for proteins involved in ubiquitin-like modifier conjugation pathway and blue for ubiquitination and proteasome degradation proteins. (C) Synapsis: Red for proteins involved in GABAergic synapses, dark blue for proteins involved in glutamatergic synapses, light green for proteins involved in dopaminergic synapses, yellow for proteins involved in the synaptic vesicle cycle, pink for proteins in the calcium signalling pathway, dark green for potassium channels, light blue for proteins in the Ras signalling pathway, orange for proteins related to circadian entrainment and purple for neurotransmitter receptors and postsynaptic transmission proteins.

**Table 1 ijms-20-03925-t001:** List of databases and software tools used in variant analysis.

**Data Bases**	**Description**	**Website**
Human mutation database (HGMD)	Database that represents an attempt to collate all known (published) gene lesions responsible for human inherited disease.	www.hgmd.cf.ac.uk/
Varsome	The human genomic variant search engine.	https://varsome.com/
GnomAD	Data from exome and genome sequencing from a variety of large-scale sequencing projects.	https://gnomad.broadinstitute.org/
dbSNP	Public-domain archive for a broad collection of simple genetic polymorphisms.	www.ncbi.nlm.nih.gov/snp/
ClinVar	Public archive of reports of the relationships among human variations and phenotypes, with supporting evidence.	www.ncbi.nlm.nih.gov/clinvar/
Specific disease databases	Databases such as RettBASE that are freely-available resources for mutation and polymorphism data pertaining to Rett syndrome and other related clinical disorders.	mecp2.chw.edu.au
**Software Tools**	**Description**	**Website**
Mutation Taster	An in silico prediction tool for the pathogenicity of a variant based on evolutionary conservation, splice-site, mRNA, protein and regulatory features.	www.mutationtaster.org/
SIFT	An in silico prediction tool for nonsynonymous variants based on sequence homology derived from closely related sequences collected through PSI-BLAST.	https://sift.bii.a-star.edu.sg/
Polyphen-2	Tool which predicts possible impact of an amino acid substitution on the structure and function of a human protein using straightforward physical and comparative considerations.	genetics.bwh.harvard.edu/pph2/
Provean	An in silico tool that predicts how nonsynonymous or in-frame indel variant will affect a protein’s biological function.	provean.jcvi.org/
Humans Splicing Finder	This tool is aimed to help studying the pre-mRNA splicing.	http://www.umd.be/HSF/

**Table 2 ijms-20-03925-t002:** List of recent publications about genetic studies in Rett syndrome (RTT) and the genes reported.

Publications	Genes
Gilissen et al. 2014 [23]	*SMC1A*
Baasch et al. 2014 [29]	*CN2A*
Saitsu et al. 2014 [30]	*TBL1XR1*
Okamoto et al. 2015 [31]	*GABRD*
Hara et al. 2015 [32]	*SHANK3*
Olson et al. 2015 [33]	*STXBP1, SCN8A, IQSEC2*
Hoffjan et al. 2016 [34]	*WDR45*
Lee et al. 2016 [35]	*SATB2*
Saez et al. 2016 [36]	*JMJD1C*
Rocha et al. 2016 [37]	*MEF2C*
Lucariello et al. 2016 [19]	*ANKRD31, CHRNA5, HCN1, SCN1A, TCF4, GRIN2B, SLC6A1, MGRN1, BTBD9, SEMA6B, AGAP6, MGRN1,VASH2, ZNF620, GRAMD1A, GABBR2, ATP8B1, HAP1, PDLIM7, SRRM3, CACNA1I*
Lopes et al. 2016 [38]	*TCF4, EEF1A2, STXBP1, ZNF238, SLC35A2, ZFX, SHROOM4, EIF2B2, RHOBTB2, SMARCA1, GABBR2, EIF4G1, HTT*
Vidal et al. 2017 [15]	*GRIN2B, GABBR2, MEF2C, STXBP1, KCNQ2, SLC2A1, TCF4, SCN2A, SYNGAP1, CACNA1I, CHRNA5, HCN1*
Sajan et al. 2017 [39]	*PWP2, SCG2, IZUMO4, XAB2, ZSCAN12, IQSEC2, FAM151A, SYNE2, SMC1A, ARHGEF10L, HDAC1, TAF1B, KCNJ10, CHD4, LRRC40, LAMB2, GRIN2B, IMPDH2, SAFB2, ACTL6B, STXBP1, TRRAP, WDR45, SLC39A13, FAT3, IQGAP3, NCOR2, GABRB2, TCF4, GRIN2A*
Allou et al. 2017 [40]	*IQSEC2, KCNA2*
Yoo et al. 2017 [41]	*GABBR2*
Vuillaume et al. 2018 [42]	*GABBR2*
Huisman et al. 2017 [43]	*SMC1A*
Wang et al. 2018 [14]	*MEF2C*
Percy et al. 2018 [44]	*CTNNB1, WDR45*
Srivastava et al. 2018 [18]	*KCNB1, IQSEC2, MEIS2, TCF4, WDR45*
Iwama et al. 2019 [45]	*ATP6V0A1, USP8, MAST3, NCOR2, WDR45, STXBP1, SHANK3, UBE3A, GABRA1, SCN2A, SCN8A, GRIN2B, IQSEC2, CAMK2B, CUX2, CACNA1D, CACNA1G, ITPR1, KIF1A, SYNGAP1, NALCN, NR2F1, IRF2BPL, MAST1, COL4A1, HDAC8, TCF4, PDHA1, PPT1, DNMT3A, MEF2C*
Schönewolf-Greulich et al. 2019 [16]	*STXBP1, SCN2A, KCNB1, TCF4, SHANK3, SMC1A*
Vidal et al. 2019 [46]	*STXBP1, TCF4, SCN2A, KCNQ2, MEF2C, SYNGAP1*

**Table 3 ijms-20-03925-t003:** List of causative genes in RTT-like diagnoses or differential diagnoses and the phenotypical overlap with Rett syndrome (modified from Schönewolf-Greulich et al., 2019 [16]).

			RTT Genes							
			*MECP2*	*CDKL5*	*FOXG1*	*STXBP1*	*TCF4*	*SCN2A*	*WDR45*	*MEF2C*
		Disorder	Rett syndrome	EEP 2	RTT, congenital variant	EEP 4	Pitt- Hopkins syndrome	EEP 11	Neurodegeneration with brain iron accumulation 5	*MEF2C* haploinsufficiency syndrome
		OMIM#	312750	300672	613454	612164	610954	613721	300894	613443
		Inheritance	XLD	XLD	AD	AD	AD	AD	XLD	AD
Present in RTT	Required	Developmental regression	+	+	+	+	-	+	+	-
	Four main criteria	Purposeful hand movements lost/absent	+	+	+	+	+	+	+	+
		Speech severe deficit/loss	+	+	+	+	+	+	+	+
		Gait abnormality	+	+	+	+	+	+	+	+
		Stereotypic hand movements	+	+	+	+	+	+	+	+
	Other common symptoms	Breathing abnormality	+	+	-	-	+	-	-	-
		ID	+	+	+	+	+	+	+	+
		Epilepsy	+	+	+	+	+	+	+	+
		Microcephaly	+	+	+	+	+	+	+	-
Not present in RTT	Exclusion criteria	CNS abnormality	-	-	+	-	+	-	+	+
	Other symptoms	Dysmorphic facial features	-	+	-	-	+	-	+	+

Abbreviations: EEP, epileptic encephalopathy; XLD, X-linked dominant; AD, autosomal dominant. Plus (+) is noted if the symptom has been described in one or more patients with a pathogenic variant in the gene. The symptoms emphasized are the main clinical features according to the 2010 classification of clinical Rett and other specific features of RTT. The gray colour indicates clinical symptoms in common with RTT.

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
