# Peer review of "Genetic Landscape of Rett Syndrome Spectrum: Improvements and Challenges"

_ijms, 2019, doi:10.3390/ijms20163925_

Round 1

Reviewer 1 Report

The manuscript by Vidal et al summarizes the recent findings on the genetic basis of Rett Syndrome and Rett-like diseases. The manuscript is timely with regards to organizing a resource for the genetic mutations associated with Rett and Rett-like disorders. I have listed my comments for the authors below:

1) The authors should address the contribution of the two MeCP2 isoforms on Rett Syndrome. This should be briefly indicated in terms of the two protein variants' expression, regulation, and relevance to the pathobiology of Rett. This is specially important, since isoform-specific partners (for example FOXG1 indicated in this review) is mainly interacting with the shorter isoform.

2) There are few articles that highlight molecular abnormalities of the brain in RTT patients. How is these highlighted genetic findings of this review (link to the affected downstream pathways) would translate from cellular model systems to human brain tissues that are the most relevant biological samples for RTT?

3) The authors should provide a critical evaluation of the genes that are listed, as to which ones would be relevant for future studies of Rett or Rett-like syndrome? 

Minor comments:

1) The sentence on page 1, lines 30-35 is too long, please break it into two sentences.

2) Please carefully read the manuscript for typo errors, and avoid abbreviating the same word (i.e. NGS) multiple times.

Reviewer 2 Report

In their manuscript, Vidal and colleagues review the existing literature on the genetic causes of Rett syndrome. It is a timely overview, since the number of genes linked to the disorder has markedly increased, specially over the last few years.

The review is systematic and well-written, and covers in a comprehensive way all reported cases of identified mutations, with an emphasys on the causative ones. In addition, functional networks that link the different genes that are typically altered in the patients are presented. This is important to illustrate the main pathways whose dysfunction leads to the disease.

Despite my overall positive evaluation of the text, some aspects need to be improved before publication:

-   The first half of Section 4 is unnecessarily long, since most of the information that is detailed in the text is present in Table 1. By contrast, more space should be given to the functional discussion of the candidate genes. Also, given the high number of patients with mutations on STXBP1 gene, a schematic diagram depicting its role in the synapse would be helpful.

-   Figure 2 needs to be provided in higher resolution.

-   Given the numerous grammar errors, the use of English language needs to be revised throughout the text. (e.g., “…the term “like” is employed in patients that the stablished clinical criteria…” (pg 2, line 21); “…sequencing allow us solve difficulties…”(pg 2, line 27); “…involved in ubiquitination processes, which UB3A is also linked to” (pg 8, line 30)).

Reviewer 3 Report

In this review Vidal et al. give a comprehensive overview of the advancements in sequencing approaches used to discover new mutations, and stress the value of such approaches in linking genetic mutations to neurodevelopmental disorders. The authors in particular focus on a number of studies describing phenotypes that while typical of RTT arise from mutations elsewhere then in MECP2 in cases that are therefore defined RTT-like syndromes. Their task is thus to highlight recurrent mutations in genes other than MECP2 that, given their association with molecular pathways commonly involving MeCP2 (i.e. epigenetic regulation, to mention one), might explain the establishment of RTT like phenotypes. The idea is that under the name of RTT spectrum disorder should be grouped conditions featuring specific phenotypes and arising from shared causative genes (“Therefore, we consider that all classical… with many causative genes”, lines 46-47 in chapter 7), a perspective that is of sure interest.

While this overview is indeed informative, some changes are required.

First (and foremost), I think the author should touch with more attention the point of the many roles MeCP2 plays and its virtual ability of regulating the expression of any given gene (Skene, 2010). The molecular pathways involving MeCP2 are thus extremely wide but, peculiarly, the transcriptional effects driven by MeCP2 are only mild in magnitude, regardless the brain area analyzed (many studies from professors Bird’s, Greenberg’s and Zhou’s labs report such peculiarity; Bedogni, 2014 give an overview on this). The fact that MeCP2 “does a lot of stuff, but only mildly” is something the authors should take into much deeper consideration.

Moreover, MECP2 is important throughout life, both during development and, later, in adulthood. This is the main topic of the recent review by Cosentino et al. on Neurosci Biobehav Rev, where the authors give a substantial re-evaluation of the so-called regression phase (please consider this and reformulate the sentence “are characterized by a period of normal development followed by a regression, and finally a slight recovery or stabilization”, chapter 2). This is another topic the authors should take into consideration.

The pathological mechanisms put in place by mutations of MECP2 (or its lack, in animal models) are thus both direct (driven by the inability of mutant MeCP2 to act properly) and indirect, as a result of compensatory mechanisms (i.e. since MeCP2 doesn’t work properly, other mechanisms are driven to survive such situation). Therefore, a number of different genes or pathways associated with RTT are not ascribable to MeCP2 itself but, rather, to “secondary effects”. Considering all the above, it is difficult to name one mechanism or gene that is preponderant compared to others. To come to my point: while the authors give quite some room to the analysis of the role of single genes in the genesis of specific phenotypes, more space, attention and discussion in the text should be given to cluster analysis linking entire molecular pathways to specific phenotypes. This I think would be a much more informative way of handling and making sense of the genetic data nowadays available. For instance, in chapter 4 “NGS results: many genes, many disorders” the description of genes from page 4 line 2 to page 5 line 34 could be easily recapitulated by transferring a substantial chunk of the information from the text to Table 1. While chapter 5 “Functions and pathways around RTT” would definitely benefit from a deeper discussion of the concept described in line 32-34 (“Focusing in MECP2 gene… downstream effects”), as some overlap in phenotypes could actually arise from common mechanisms leading to developmental derangements rather than from the mutations of single genes involved in specific pathways. Such developmental derangement could be “shared” simply because arising from similar compensatory mechanisms. Please discuss this point. After all, Mecp2 is expressed as early as the stage of during embryonic brain development (Baj, 2014; Mellios, 2018; Cobolli Gigli, 2018; Feldman, 2016; Tang, 2016; Li, 2013). Accordingly, the early, pre-symptomatic effects of its lack (or mutations) are now solid; see Cosentino, 2019 again for a clear (or, rather, clearer) picture. For instance, many of the 21 genes mentioned in chapter 5 (“involved in synaptic function”) are downregulated in E15 Mecp2 null embryonic cortexes, when no synaptic contacts are yet formed (Bedogni, 2016). To conclude, I think evaluating the data described in this review from a perspective that take into account “time” as a variable for the establishment of specific phenotypes could add more insights and value to this study. This is my second “major” suggestion.

Please also consider these points:

1: Chapter 3: “a substantial increase in the identification of new diseases or genes …”, could you explain in what way these new techniques can “identify new diseases”?

2: Chapter 6: “may also derive from an improved understanding of the polygenic RTT-like phenotype”, I’m not sure what the authors mean with this.

3: Chapter 3: the sentence in chapter 3 “Using variation databases, such as ExAC (Exome Aggregation Consortium; exac.broadinstitute.org/), gnomAD (Genome Aggregation Database; https://gnomad.broadinstitute.org/), LOVD (Leiden Open Variation Database; www.lovd.nl) or HGMD (Human Genome Mutation Database; http://www.hgmd.cf.ac.uk) and software tools like SIFT (Sorting Intolerant From Tolerant; sift.bii.a-star.edu.sg), HSF (Human Splicing Finder; umd.be/HSF/) or PolyPhen-2 (Polymorphism Phenotyping v2; http://genetics.bwh.harvard.edu/pph2 )” should be part of an additional table.

Eventually, a thorough and detailed editing is needed. The construction of a number of sentences is intricate or obscure and requires revision.

Here are some examples, but more things could be polished up:

Introduction: “that encodes a chromatin-associated protein that can both activate and repress transcription and is required for the maturation of neurons and is developmentally regulated”, too many information in the same sentence.

Introduction: “described by the doctor Andreas Rett in 1966”, get rid of “the”.

Introduction: “In this variant RTT group of patients”, reformulate.

Introduction: “in the huge heterogeneous group”, in guess it’s rather “hugely heterogeneous”.

Chapter 2: “The standard diagnosis criteria used to establish the clinical diagnosis of RTT”, reformulate.

Chapter 2: “The classical RTT consists in four main criteria that patients should have present”, reformulate.

Chapter 2: “there are described some features”, reformulate.

Chapter 2: “the term “like” is employed in patients that the stablished clinical criteria are not fulfilled”, amend.

Chapter 2: “respecting the clinical diagnosis”, obscure.

Chapter 3: “Targeted panels that focus on individual genes, specific regions of interest or on a subset of genes associated with a wide variety of inherited disorders and are usually the first line of testing, while WES is reserved for cases in which targeted testing has been uninformative”, I believe and is a typo, but the sentence is too long anyway.

Chapter 3: “DS patients with mosaicism have milder phenotypes than the those without mosaicism present”, reformulate.

Chapter 4: “the number of positive results has been increased in genes”, reformulate.

Chapter 4: “are the most common genes related to an RTT-like phenotype”, reformulate.

Chapter 5: “Focusing in MECP2 gene, which performs many tasks during the neurodevelopment such as regulating the gene expression of other genes, modulating epigenetic imprinting and neurotransmitter actions, makes it challenging to gather information about the downstream effects”, reformulate.

Chapter 5: “Integrating transcription factors they identified a possible link how MECP2 regulates cytoskeleton organisation via MEF2C and CAPG”, reformulate.

Chapter 6: “therapies must focus on personalized treatments for each individual, depending on which gene and which type of mutation carries”, “carries” I believe refers to “individual”, reformulate.

Chapter 7: “New candidate genes that functional studies are necessary to establish their potential role in this disease pathogenesis”, obscure, reformulate.

Round 2

Reviewer 1 Report

The manuscript has improved but few minor comments remain:

1- The authors have acknowledged the MECP2E1-specific mutations are found in RTT patients, but the contribution of the two isoforms in mouse models (Mecp2e1- and Mecp2e2-deficient mice) should also be highlighted. This is important for validation of RTT mouse models, in terms of RTT pathobiology. 

2- It has been shown that MeCP2E1 is the dominant protein isoform in the brain, thus it is shown experimentally not suggested, as indicated in line 1 of page 2. Please revise accordingly and reference properly. In contrast, MeCP2 shows a more region-specific distribution in mouse brain. Please revise accordingly and reference properly.

3- On page 8, the name of the two genes on line 25 should be in Italics: Mef2c and Scn2a.

4- I appreciate the efforts that the authors have made to improve the manuscript, but one aspect that is not properly addressed is how the cell model systems with Mecp2/MECP2-deficiency are conserved from mice to humans. In this regards, the authors should address the pathway deficiencies that are found in the human neurons (in vitro) and in human RTT brain (in vivo from patients). There are important literature in this regards in both human neurons and RTT-associated brain tissues of the patients. This information are very important for completion of this review and should be included.

Reviewer 3 Report

The authors answered many question and the manuscript is ameliorated in terms of scientific contents. However, despite the suggestion of a major grammar revision, many amendments are still required, as many sentences are still cryptic. In my opinion this article is still not in a publishable form.

Here are my points:

“This gene encodes a chromatin-associated protein that presences the methyl-CpG binding domain and can activate and repress transcription; it is essential for the maturation of neurons and normal function of nerve cells. As a result, this gene encodes a chromatin-associated protein that can both activate and repress transcription and is required for the maturation of neurons and is developmentally regulated.”, reformulate and get rid of duplicated concepts.

“Therefore, the Sanger sequencing of the MECP2 gene was included”, why “therefore”?

“The diagnosis criteria used in RTT are clinical criteria; there is no biomarker—it was first published in 1966 and revised in 2002 and 2010 (7).” Reformulate/amend.

“Followed by slight recovery or pseudo-stationary stage and finally in the adult age worsening of motor performance and autonomic dysfunctions can appear.” Reformulate, unclear

“Otherwise some atypical forms congenital and early seizure shows developmental impairment/delay since the first months of life.” reformulate

“is used in patients which not fulfilled the stablished clinical criteria, but present an overlapping phenotype with the disease.” “In patients that do not fulfill”, replace “stablished” with “established”

“whole exome/genome sequencing allow us to overcome difficulties”, replace “allow” with “allows”

“has yet to be stablished”, replace “stablished” with “established”

“diagnostic rate in the hugely heterogeneous group”, replace “the” with “this”

“identified as causative for an RTT or RTT-like”, replace “an” with “a”

“The STXBP1  gene (syntaxin-binding protein 1; OMIM#602926) encoded a transmembrane”, replace “encoded” with “encodes”

“Pathogenic variants in this gene that produce a reduction of its expression has been demostrate an increase synaptic depression in GABAergic and glutamatergic synapses, but with higher impact on GABAergic interneurons (44), a process that has been shown to be disruptive in RTT patients (45).” Please amend.

“one of the four main criteria, is absent in PHS.”, I believe it’s PTHS.

“formation (63). And WDR45  cOK mice”, get rid of “And”.

“evaluation of these genes and its linking to RTT should be done”, “replace and its lining” with “and their link”.

“In order to relate a gene with RTT have to be clear about several points”, obscure.

“A well clinical characterizations of the patients. 2) A well comprehension of the functions of the candidate gene.”, “a well clinical” and “a well comprehension” replace “well” with a different adjective.

“A strongly evidence that mutation found it is pathogenic.”, obscure.

“However, using REACTOME, an open-source, open access, manually curated and peer-reviewed pathway database (https://reactome.org ), and STRING, a database of known and predicted protein–protein interactions (https://string-db.org/ ), we can identify some groups of genes with functions involved in common mechanisms which are altered in RTT/RTT-like patients, such as chromatin modulation, synaptic function and ubiquitin conjugation (Figure 2).”, split this long sentence in shorter ones.

“For example, altered calcium homeostasis seems to be responsible for an abnormal neuronal development and generates epilepsy; tubulin, ERM and MEF2C are some of the altered proteins related to cytoskeletal abnormalities that are present, not only in RTT but also in Angelman syndrome (65); cholesterol biosynthesis is altered in RTT and in Smith-Lemli-Opitz syndrome, in which it has been pointed out to be the cause of the autism and malformations (67); the NF-kB pathway, which is involved in nervous system development, synaptic transmission and cognition, is altered in RTT and RTT-like patients and seems to be the cause of mental retardation (68); the neurotransmitter imbalance of GABA, Glutamine and Dopamine has been observed in patients with RTT, autism and Parkinson disease (PD)”, too many infos in one sentence, split it in shorter ones.

“Multiple evidences are increasing and emerging that with a perfect understanding of these mechanisms it is very important for be able to select the most appropriate treatment for all of these similar but not identical patients”, very very very cryptic.

“For instance, a treatment which targets neuronal maturational defects seen in MECP2 mutations may not be effective for a disruption of GABAergic, glutamatergic or dopaminergic synapses. In this way, therapies must focus on personalized treatments for each individual, depending on which gene and which type of mutation is present.”, quite cryptic.

“the new genes recently associated with RTT-like phenotypes, without a clear description of their biological function network, need to be elucidated”, replace “need to be elucidated” with “need further studies”.

Round 3

Reviewer 3 Report

I hope the authors will not mind, but again some stuff should be amended:

1: "Pathogenic variants in this gene produce a reduction of its expression has been shown an increase synaptic depression in GABAergic and glutamatergic synapses, but with higher impact on GABAergic interneurons (48)", please replace with: "Pathogenic variants in this gene reducing its expression depress the functions of GABAergic and glutamatergic synapses, particularly in GABAergic interneurons".

2: "At the moment, more than 80 genes have been associated with RTT/RTT-like. But, a critical evaluation of these genes and their link to RTT should be done. In order to relate a gene with RTT several points have to take into consideration:", replace with "At the moment , more than 80 genes have been associated with RTT/RTT-like, but their link with RTT should be critically evaluated. In order to relate a gene with RTT several points have to be taken into consideration:"

3: "Moreover, it has been stablished that MeCP2 regulates KCC2 expression through inhibiting RE1-silencing transcriptional factor", replace "stablished" with "established".

4: "It has been hypothesized that enhancing KCC2 expression will could increase the efficacy of GABAergic inhibition", either "will" or "could", i would go for "could".

Since these 4 points need more attention anyway, i think the authors should consider/mention these very recently published articles: Tang X, 2019 (PMID:31366578) and Lozovaya N, 2019 (PMID:31239460) as both considering the pathogenic role of diminished Kcc2 expression in Mecp2 null models.
